# Linearizing Large Language Models

**Jean Mercat**[*]  **Igor Vasiljevic**[*]  **Sedrick Keh**[*]
**Kushal Arora**    **Achal Dave**    **Adrien Gaidon**    **Thomas Kollar**

Toyota Research Institute
{jean.mercat, igor.vasiljevic, sedrick.keh}@tri.global
{kushal.arora, achal.dave, adrien.gaidon, thomas.kollar}@tri.global

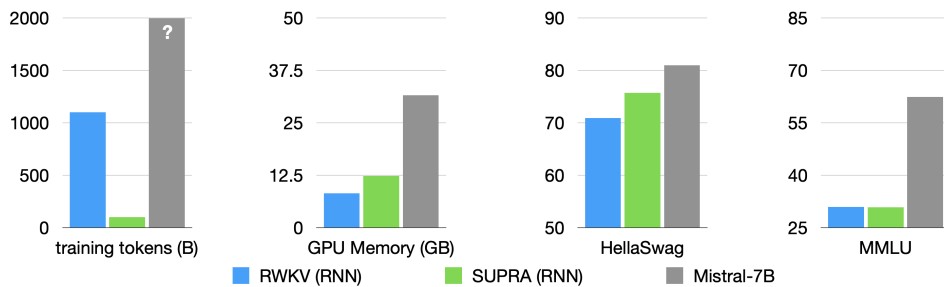

Figure 1: We reuse pre-trained LLMs (gray) and convert them to RNNs with minimal uptraining (SUPRA), outperforming linear attention models (RWKV) on natural language tasks like HellaSwag, attaining the same memory advantages (center), but also inheriting the limitations of RNNs on tasks like MMLU (right).

## Abstract

Linear transformers have emerged as a subquadratic-time alternative to softmax attention and have garnered significant interest due to their fixed-size recurrent state that lowers inference cost. However, their original formulation suffers from poor scaling and underperforms compute-matched transformers. Recent linear models such as RWKV and Mamba have attempted to address these shortcomings by proposing novel time-mixing and gating architectures, but pre-training large language models requires significant data and compute investments. Thus, the search for subquadratic architectures is limited by the availability of compute and quality pre-training datasets. As a cost-effective alternative to pre-training linear transformers, we propose Scalable UPtraining for Recurrent Attention (SUPRA).[1] We present a method to *uptrain* existing large pre-trained transformers into Recurrent Neural Networks (RNNs) with a modest compute budget. This allows us to leverage the strong pre-training data and performance of existing transformer LLMs, while requiring 5% of the training cost. We find that our linearization technique leads to competitive performance on standard benchmarks, but we identify persistent in-context learning and long-context modeling shortfalls for even the largest linear models. Our code and models can be found at https://github.com/TRI-ML/linear_open_lm.

---

[*]Equal contribution.

[1]We borrow the term "uptraining" from Ainslie et al. (2023) to refer to continued training with a modified architecture, as opposed to fine-tuning, which usually refers to continued training on a different dataset.

# 1 Introduction

Over the last few years, Transformers (Vaswani et al., 2017) have displaced Recurrent Neural Networks (RNNs) in sequence modeling tasks, owing to their highly parallel training efficiency and unmatched scaling performance (Kaplan et al., 2020). However, this training efficiency comes at the cost of inference cost that scales linearly with the number of tokens, compared to the fixed-cost inference of RNNs. The memory-intensive nature of transformers has led to renewed interest in recurrence—the fixed-size hidden state remains an attractive modeling proposition to reduce the cost of inference for language and multimodal models.

Several recent works, starting with *Linear Transformers* (Katharopoulos et al., 2020), have observed a relationship between a linearized form of attention and recurrence, leading to a duality between transformers and RNNs: models can be trained with sequence parallelism (i.e. as transformers, avoiding backpropagation through time), but can operate as RNNs at inference time. Although this architecture allows efficient training of RNNs, softmax transformers continue to outperform linear transformers across natural language understanding benchmarks. A number of novel RNN architectures have attempted to bridge this performance gap. These include RWKV (Peng et al., 2023a), Retentive Networks (Sun et al., 2023), TransNormer (Qin et al., 2022a), and more recently, Griffin (De et al., 2024) and RecurrentGemma (Griffin Team et al., 2024). These models are pre-trained on the same pre-training datasets as transformers and show promising results.

State-space models (Gu et al., 2021) (SSMs) are another recurrent alternative to softmax transformers, combining RNNs and convolutional networks to efficiently model long sequences. The Mamba (Gu & Dao, 2023) architecture is a SSM that shows impressive performance at smaller scales, matching or exceeding the performance of softmax transformers on a number of natural language understanding (NLU) benchmarks. However, a gap remains for long-context NLU tasks, showing a persistent advantage of softmax attention.

Architecture search at the scale of large language models is expensive. Rather than pre-training linear models, another approach is to *convert* an existing transformer into an RNN; Kasai et al. (2021) proposed to uptrain encoder-decoder transformers into RNNs by introducing an approximating MLP attention module. Zhang et al. (2024) improved on this method by adding a loss to match softmax attention to approximate more closely the base transformer.

While approximating attention is an intriguing approach to re-using pre-trained transformers, it leads to instability and poor performance when uptraining large-scale models. We instead take a different approach: rather than *approximate* softmax attention, we *replace* it with a linear kernel and a normalization strategy to uptrain the most performant LLMs into RNNs (see Figure 2). We take advantage of models trained on high-quality, proprietary datasets for trillions of tokens (e.g. Mistral (Jiang et al., 2023) and Llama2 (Touvron et al., 2023)). Fine-tuning these models on publicly available data for a small fraction of pre-training tokens (see Figure 1), we obtain linear models that are competitive with the best linear transformers for a fraction of the compute. We call our approach Scalable UPtraining for Recurrent Attention (SUPRA).

Our contributions are as follows:

- We propose Scalable UPtraining for Recurrent Attention (SUPRA), a linearization strategy to uptrain state-of-the-art LLMs into performant RNNs.
- We show that this simple uptraining technique is competitive with the strongest pre-trained recurrent LLMs.
- We investigate the limitations of recurrent LLMs, comparing pre-trained and uptrained RNNs to transformers, revealing a persistent gap for in-context learning and long-context tasks.

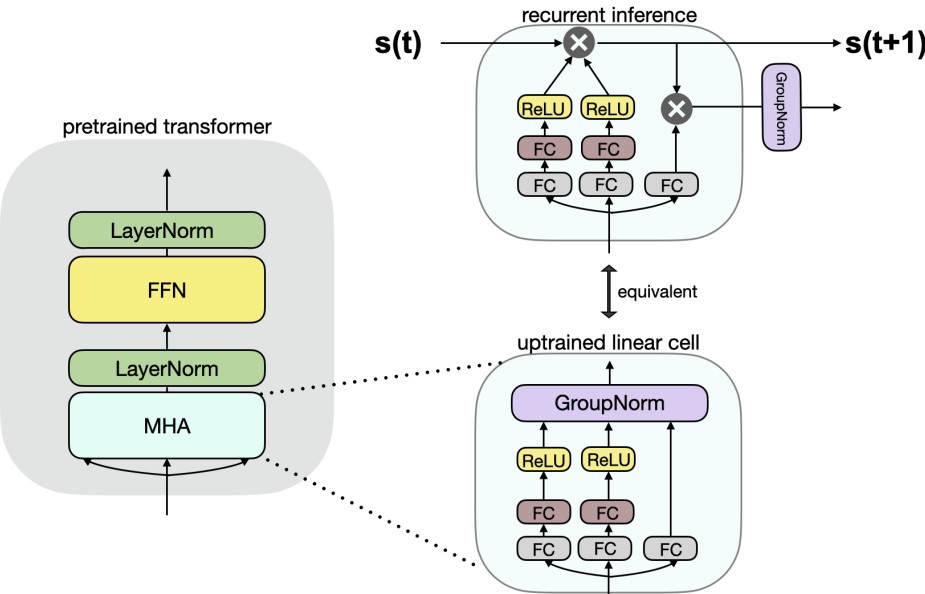

Figure 2: Our linearization strategy: we replace the softmax normalization with GroupNorm (GN) and introduce a small MLP to project the queries and keys, converting a pre-trained attention block (left) to a linear attention (right). The model can be be trained in parallel as a transformer and used recurrently at inference time with a mathematically equivalent reformulation.

## 2 Methodology

In this section we review linear transformers, and describe the linearization technique of Kasai et al. (2021) as it lays the groundwork for our approach. Finally, we present SUPRA, our method for uptraining large transformers into RNNs.

### 2.1 Background: Linear Attention

*Linear Transformers* (Katharopoulos et al., 2020) establish a connection between transformers and RNNs, generalizing the definition of attention by replacing the softmax dot-product attention $\mathbf{v}'$ with a more generic similarity function $\text{sim}(\mathbf{q}, \mathbf{k})$ between the queries $\mathbf{q}$ and keys $\mathbf{k}$:

$$\mathbf{v}'_i = \frac{\sum_{j=1}^i \text{sim}(\mathbf{q}_i, \mathbf{k}_j)\mathbf{v}_j}{\sum_{j=1}^i \text{sim}(\mathbf{q}_i, \mathbf{k}_j)}. \tag{1}$$

Standard softmax attention is a special case, using $\text{sim}(\mathbf{q}, \mathbf{k}) = \exp\left(\frac{\mathbf{q}^T\mathbf{k}}{\sqrt{d}}\right)$.

The authors explore several alternative functions for $\text{sim}(\mathbf{q}, \mathbf{k})$, including a linear kernel. Their main architecture uses the similarity function $\text{sim}(\mathbf{q}, \mathbf{k}) = \phi(\mathbf{q}) \cdot \phi(\mathbf{k})$ with a fixed exponential linear unit kernel $\phi(x) = \text{elu}(x) + 1$. They show the computational benefits of linear attention and, more importantly for this work, they demonstrate how such a model can be expressed as an RNN in the case of attention with causal masking.

**Recurrent Inference** Linear attention can be expressed as an RNN that updates a state $\mathbf{s}_i$ and a normalization factor $\mathbf{z}_i$ at each time step. Katharopoulos et al. (2020) call these terms the *attention memory* and *normalized memory*. This RNN formulation is mathematically

equivalent to linear attention, allowing the user to choose the most efficient one for a given task and hardware. Consider a stream of tokens we want to generate $X = [\mathbf{x}_1, \mathbf{x}_2, \mathbf{x}_3, \ldots]$. At inference time, we use the following update rule, where subscripts denote timestep in the recurrence (calling $\mathbf{k}_i = W_K \mathbf{x}_i$, etc):

$$\mathbf{s}_0 = 0 \quad \mathbf{z}_0 = 0 \tag{2}$$

$$\mathbf{s}_i = \mathbf{s}_{i-1} + \phi(\mathbf{k}_i)\mathbf{v}_i^T \tag{3}$$

$$\mathbf{z}_i = \mathbf{z}_{i-1} + \phi(\mathbf{k}_i) \tag{4}$$

$$\mathbf{v}'_i = \frac{\phi(\mathbf{q}_i)^T \mathbf{s}_i}{\phi(\mathbf{q}_i)^T \mathbf{z}_i} \tag{5}$$

The state $\mathbf{s}_i$ acts as a constant-size KV cache. Instead of appending new values to the cache, the state is updated. This allows for inference cost that is constant in the number of generated tokens.

## 2.2 Finetuning a Transformer into an RNN

Kasai et al. (2021) introduced a linear transformer uptraining procedure that converts a pre-trained softmax transformer into an RNN by *approximating* the attention computation with multi-layer perceptrons (MLPs). The method (T2R) starts with a softmax attention model, and linearizes the softmax operation. Recall the kernel linear attention similarity function:

$$\text{sim}(\mathbf{x}, \mathbf{y}) = \phi(\mathbf{x}) \cdot \phi(\mathbf{y}). \tag{6}$$

Instead of choosing $\phi$ as a simple non-linearity, the authors use a trainable layer:

$$\phi(\mathbf{x}) = \text{relu}(W\mathbf{x} + \mathbf{b}). \tag{7}$$

The weights are shared between keys and queries for a given attention head. By using $\phi$ and rearranging the operations, attention can be written as:

$$\mathbf{v}'_i = \frac{\phi(\mathbf{q}_i)^T \sum_{j=1}^{i} \phi(\mathbf{k}_j)\mathbf{v}_j^T}{\phi(\mathbf{q}_i)^T \sum_{j=1}^{i} \phi(\mathbf{k}_j)}. \tag{8}$$

This allows the recurrent inference described in Section 2.1. However, this formulation has a number of drawbacks. First, it requires a significant re-training of the model, using approximately 20% of pre-training tokens for conversion, while suffering a $5 - 10\%$ drop in performance on language benchmarks. Furthermore, this approach was tested on relatively small models ($\approx 100M$ scale). Because it mimics the attention formulation closely, it suffers from stability issues at larger scales. To address these issues, we modify the approach to adapt it to large-scale model uptraining.

## 2.3 SUPRA: Scalable UPtraining for Recurrent Attention

Rather than pre-training linear models from scratch, we choose to instead *uptrain* state-of-the-art transformers. Leveraging models that take advantage of high-quality (but proprietary) pre-training datasets, we linearize them using a modest fraction of pre-training data (see Figure 1). We build on T2R, identifying two major issues and proposing SUPRA, an approach to fine-tuning very large transformers into RNNs.

We first follow the literature in linear transformers and identify the **normalization factor** in linear attention as unstable (e.g. TransNormer (Qin et al., 2022a)). In Section 3.3 we show that uptraining a 1B model following the procedure in T2R causes a large drop in performance. We instead follow Retentive Networks (Sun et al., 2023) and replace the normalization with a GroupNorm operation.

Next we note that linear attention suffers more with absolute positional encoding than softmax attention, and a modern relative positional encoding scheme like RoPE (Su et al., 2021) is crucial for competitive performance. Rather than training a linear transformer from scratch incorporating these findings (RetNet, TransNormer) we use MLP kernels to *convert* large language models into RNNs.

Starting with the pre-trained model, we add weights shared between keys and queries $W$, $\phi(\mathbf{x}) = \text{relu}(W\mathbf{x} + \mathbf{b})$ and use the rotary positional embedding (RoPE (Su et al., 2021)) such that the similarity function becomes

$$\text{sim}(\mathbf{q}_i, \mathbf{k}_j) = \text{RoPE}(\phi(\mathbf{q}_i)) \cdot \text{RoPE}(\phi(\mathbf{k}_j)). \tag{9}$$

We normalize the output with a GroupNorm (Wu & He, 2018) instead of dividing by the sum of $\text{sim}(\mathbf{q}_i, \mathbf{k}_j)$ (as in T2R). We use a fixed decay vector $\gamma \in (0,1)^h$, with $h$ heads, as in Sun et al. (2023). This leads to the following attention formulation (see Figure 2 for a graphical representation):

$$\mathbf{v}'_i = \text{GroupNorm}\left(\sum_{j=1}^{i} \gamma^{i-j}\text{sim}(\mathbf{q}_i, \mathbf{k}_j)\mathbf{v}_j\right). \tag{10}$$

These new parameters are trained jointly with the rest of the network; at test time, we use the recurrent formulation for inference.

## 3 Experiments

We uptrain a variety of models from the 1B to 7B range into RNNs (Llama2 (Touvron et al., 2023) and Mistral (Jiang et al., 2023)), and evaluate our models in two settings: standard language understanding benchmarks and long-context evaluations. We compare the results of different architectural choices and training strategies, and then show the limitations of linear models on various benchmarks, describing the persistent gap between vanilla attention and recurrence. We choose Llama2-7B and Mistral-7B as our base models for uptraining, but our recipe is general to any transformer model.

| Model | Size | Tokens | HellaSwag | PIQA | WG | ARC-E | ARC-C | MMLU | Average |
|---|---|---|---|---|---|---|---|---|---|
| StableLM2 | 1.6B | 2000 | 69.0 | 76.7 | 63.6 | 68.6 | 38.9 | 38.4 | 59.2 |
| StableLM | 3B | 1000 | 73.8 | 79.3 | 65.8 | 72.1 | 40.0 | 44.2 | 62.5 |
| Gemma | 2B | 2000 | 71.4 | 78.6 | 64.4 | 74.0 | 41.5 | 41.2 | 61.9 |
| Mamba | 1.4B | 600 | 59.0 | 73.9 | 61.4 | 65.5 | 32.9 | 25.2 | 53.0 |
| RWKV-5 | 1.5B | 1100 | 53.1 | 71.6 | 59.0 | 62.2 | 32.7 | 26.2 | 50.8 |
| Mamba | 2.8B | 600 | **66.2** | **75.8** | **63.4** | **69.7** | **36.3** | **26.3** | **56.3** |
| Llama2 | 7B | 2000 | 76.0 | 79.1 | 69.1 | 76.3 | 46.3 | 45.9 | 65.4 |
| Llama2 | 7B | +20 | 76.3 | 78.8 | 70.1 | 76.8 | 46.2 | 43.8 | 65.3 |
| Gemma | 7B | 6000 | 80.7 | 81.9 | 73.7 | **81.1** | 53.2 | **62.9** | 72.2 |
| Mistral | 7B | 8000(?) | **81.0** | **82.1** | **74.0** | 80.9 | **53.8** | 62.4 | **72.4** |
| RetNet | 6.7B | 200 | 60.7 | 75.4 | 58.1 | – | – | – | – |
| RWKV-5 | 7B | 1100 | 70.9 | 77.2 | 67.4 | 71.8 | 43.6 | 31.0 | 60.3 |
| RWKV-5-1.7T | 7B | 1700 | 73.0 | 78.6 | **72.9** | 75.8 | 45.6 | **34.9** | 63.5 |
| Mamba (ours) | 7B | 1200 | **77.9** | **81.0** | 71.8 | **77.5** | **46.7** | 33.3 | **64.7** |
| Llama2-SUPRA | 7B | +20 | 71.8 | 78.6 | 65.8 | 71.1 | 39.5 | 24.9 | 58.6 |
| Mistral-SUPRA | 7B | +20 | 74.8 | 80.1 | 67.4 | 74.6 | 42.3 | 28.0 | 61.2 |
| Mistral-SUPRA | 7B | +100 | 77.1 | 80.4 | 70.3 | 75.9 | 45.8 | 34.2 | 64.0 |

Table 1: Linear models (RNNs and SSMs) highlighted in gray. 5-shot results are used for MMLU. Norm results are used for PIQA, HellaSwag, ARC-C. RetNet results taken from RetNet paper.

We compare our procedure to a variety of pre-trained recurrent models. Given that the largest available state-space models are at the 2.8B scale, we also train a Mamba model on the RefinedWeb (Penedo et al., 2023) dataset from scratch for 1.2T tokens, to serve as a strong baseline for a pre-trained recurrent model [2].

We use a fork of OpenLM (Gururangan et al., 2023) for all training and fine-tuning. Please see Section 7 for hyperparameters and further details on reproducibility.

**Language Modeling.** In Table 1 we report results on standard NLU evaluations using the Eleuther evaluation harness (Gao et al., 2023). We primarily compare to transformers and linear models at the 7B scale, and we train a Mamba model at 7B for comparison with RWKV-5. As our model is initialized from strong pre-trained transformers (Llama2 and Mistral-7B), it preserves performance on most benchmarks (except MMLU; see Section 4 for a discussion below). Our technique outperforms RWKV-5 with minimal uptraining and is competitive with our 7B Mamba trained from scratch on 1.2T tokens.

| Model | Size | Train Context | Qasper (2-shot) | | | | NarrativeQA (0-shot) | | | |
|---|---|---|---|---|---|---|---|---|---|---|
| | | | 2048 | 4096 | 8192 | 16384 | 2048 | 4096 | 8192 | 16384 |
| Llama1-7B | 7B | 2048 | 24.43 | 7.23 | 5.08 | 4.88 | 21.44 | 1.03 | 0.0 | 0.0 |
| Llama2-7B | 7B | 4096 | 23.26 | 27.26 | 6.20 | 5.49 | 21.32 | 22.61 | 0.0 | 0.0 |
| Llama2-7B* | 7B | 4096 | 23.26 | 27.26 | 31.46 | 25.52 | 21.32 | 22.61 | 23.0 | 14.27 |
| Mistral-7B | 7B | 8196 | 21.53 | 25.50 | 33.61 | 6.88 | 24.94 | 26.90 | 25.93 | 0.63 |
| RecurrentGemma-2B | 2.7B | 8192 | 22.44 | 13.16 | 13.42 | 12.66 | 19.80 | 11.59 | 12.93 | 12.95 |
| RWKV-5-1.7T | 7B | 2048 | 22.28 | 23.87 | 22.30 | 20.35 | 17.77 | 18.65 | 17.81 | 16.00 |
| Mamba (ours) | 7B | 2048 | 19.68 | 5.58 | 5.90 | 6.32 | 19.70 | 0.28 | 0.0 | 0.0 |
| Mistral-SUPRA | 7B | 2048 | 19.44 | 17.13 | 17.11 | 17.22 | 18.99 | 17.76 | 17.75 | 17.74 |

Table 2: Long context evaluations. Performance at various context size cutoffs for Qasper (2-shot) and NarrativeQA (0-shot). * denotes linear RoPE scaling with YaRN (Peng et al., 2023b).

**Long Context.** Recurrent models were thought to perform well on long-context tasks because of their ability to preserve performance beyond their training sequence size. However, their downstream performance on long-context tasks has not been well-documented. Prior studies either do not conduct long-context evaluations (Katharopoulos et al., 2020; Kasai et al., 2021), evaluate only on perplexity (Sun et al., 2023; De et al., 2024; Gu & Dao, 2023), or evaluate on datasets which require task-specific training (Peng et al., 2023a). Instead, we consider downstream *natural language* tasks from the SCROLLS benchmark (Shaham et al., 2022a). Specifically, in Table 2 we present two tasks – Qasper (Dasigi et al., 2021) and NarrativeQA (Kočiský et al., 2018) – from the set of tasks evaluated in the Llama2-Long report (Xiong et al., 2023). We evaluate both tasks with an input context cut-off at different lengths. A strong long-context model should perform better given more context. However, the training context lengths for these models do not go beyond 8k tokens. Transformer models show the strongest results up to the context length they were trained for but degrade beyond that. Interestingly, applying the YaRN trick (Peng et al., 2023b) enables transformers to scale beyond their training context quite well. RWKV shows a strong ability to handle much longer context than its training. Our Mamba model on the contrary is not able to generalize beyond its training context length. Surprisingly, the RecurrentGemma model (Griffin Team et al., 2024) shows degrading performances even within its training context length. Finally, our Mistral-SUPRA model preserves some performance at larger context lengths but we believe it to result from the decay along the context length that shortens the effective context. This is discussed in more details below. We find a significant gap in performance between transformers and available linear models, including models uptrained from strong long-context transformers. We speculate that more sophisticated

---

[2]The Mistral-SUPRA and Mamba-7B models are released along with the code.

recurrent state update rules may be required to perform well at this task. Ideas such as gating strategies (De et al., 2024), higher order linear attention (Mercat, 2020), or associative binding (Munkhdalai et al., 2024) could be explored.

**Decay factors.** The default decay factors proposed in Qin et al. (2024a) gives better results than no decay on short context benchmarks but at a long range, the decay cancels out the influence of the context $(\max(\gamma)^{2048} = 3.35e^{-4})$. This can be related to a smooth version of window attention (Beltagy et al., 2020). However, the figure 3 shows that the cross entropy of the linear model plateaus at longer range while the original model keeps improving. This is also confirmed on downstream evaluation, in table 2. As more context is given to the model, the long-range evaluation performance plateau. When using the decay values proposed in Sun et al. (2023), that allow longer range attention, we observe a performance drop on short-context benchmarks and no substantial improvement on long-context evaluation.

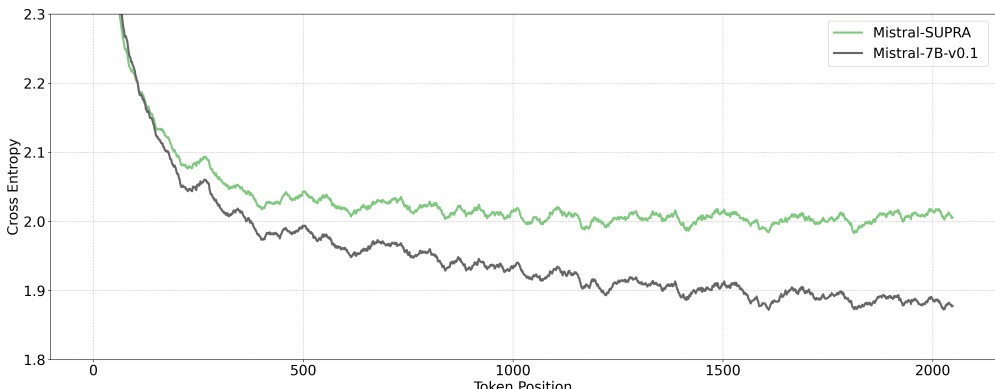

Figure 3: Plot of the cross entropy results for the base Mistral model and our Mistral-SUPRA model for each token positions in the sequence. A subset of the C4 dataset was used. The transformer model benefits from more context while the linear model plateaus.

| Model | Size | Tokens | HellaSwag | ARC-E | ARC-C |
|---|---|---|---|---|---|
| Mamba | 1B | 100B | 58.3 | 62.7 | 29.1 |
| SUPRA (from scratch) | 1B | 100B | 54.7 | 59.6 | 27.8 |
| T2R (from scratch) | 1B | 100B | 55.2 | 61.0 | 28.4 |
| Transformer | 1B | 100B | 55.9 | 60.4 | 29.3 |
| Transformer* | 1B | 1.6T | 62.1 | 66.2 | 34.3 |
| Fine-tune only new weights | 1B | (1.6T)+10B | 33.2 | 37.8 | 22.6 |
| 2-step fine-tune | 1B | (1.6T)+10B+10B | 56.1 | 62.2 | 30.4 |
| T2R | 1B | (1.6T)+10B | 40.6 | 53.8 | 27.9 |
| SUPRA | 1B | (1.6T)+10B | 57.0 | 62.4 | 31.6 |
| SUPRA-no activation | 1B | (1.6T)+10B | 55.9 | 61.6 | 29.8 |
| SUPRA-no embed-no activation | 1B | (1.6T)+10B | 36.5 | 48.5 | 23.7 |
| SUPRA-1+elu | 1B | (1.6T)+10B | 56.5 | 64.0 | 31.1 |
| SUPRA-no embed-1+elu | 1B | (1.6T)+10B | 49.9 | 57.4 | 30.6 |
| SUPRA-weaker base | 1B | (100B)+10B | 51.7 | 57.5 | 27.4 |

Table 3: Ablating different choices for linear uptraining: note the importance of normalization. For the second half of the table, we uptrain a transformer trained on 1.6T tokens on 10B further tokens. *This model was trained on a different mix of data.

**Ablations.** Table 3 compares transformers pre-trained on 100B tokens to Mamba (Gu & Dao, 2023), T2R (Kasai et al., 2021), and our approach. At this scale, with 100B tokens of training, the Mamba model performs best and other models show similar performance. The second half of Table 3 shows results for uptraining from a pre-trained transformer. TheT2R (Kasai et al., 2021) uptraining was unstable, yielding poor results compapred to SUPRA. This confirms that normalization is key for maintaining performance of the base LLM when uptraining.

To test the hypothesis that linear attention approximates the softmax attention, we experimented with a 2-step approach. The first step trains only the new parameters such that the model could learn to approximate the softmax. The second step fine-tunes all the weights. The results show no benefit from the two steps approach and indicates that the softmax is not approximated. See Appendix A for a different approach to compare softmax attention and linear attention.

We also experiment with different kernel choices. The "no embed" models remove the additional fully connected layer that we added. Removing this layer significantly degrades the results. The "1+elu" models replace the ReLU activation with $x- > 1 + \text{elu}(x)$ that is used in Katharopoulos et al. (2020) and that is simply a smooth version of the ReLU function. The "no activation" models do not use any activation function. It seems that the bias added by the linear layer of the kernel is very important to produce good results but the activation itself has little impact on the performance. Results are slightly better with a non-linear activation than without.

Finally, we compare the results of SUPRA uptrainings from two pre-trained softmax models. It appears that pre-training a linear model for a 100B token yields better results than fine-tuning a softmax model that was trained with the same budget. These results also shows, along with the comparison of LLama2-SUPRA and Mistral-SUPRA in Table 1, that SUPRA benefits significantly from a stronger pre-trained transformer. Thus, given a limited training budget, using SUPRA from a strong pre-trained model is the best option.

## 4 Discussion

**Comparison to pre-training SSMs/RNNs.** With only 20B tokens of training, which represents $2 - 10\%$ of RWKV and RetNet training cost, we obtain a model that outperforms both on HellaSwag and that is competitive on other benchmarks (see Table 1). Given the existing performance gap between the strongest transformer models and the most performant linear models, SUPRA is a simple recipe for conversion, allowing the study of strong RNNs with limited uptraining.

**Comparison to Transformers on Short-Context Tasks.** Our approach does not explicitly approximate attention from the base transformer model (see Appendix A), we do see a modest drop in performance across all benchmarks compared to softmax transformers. This could be partially explained by the lower quality of our data compared to the pre-training mix used to train models like Mistral-7B. It is also likely that linear transformers are inherently less expressive. However, the performance drop is relatively modest on most benchmarks, and significantly smaller than the drop from T2R uptraining, which shows the relevance of our approach.

**Long Context Comparisons.** Prior work on linear attention showcased similar or better validation set perplexity to transformer models over long context (e.g. Sun et al. (2023)) but did not evaluate linear models on *natural language* long-context evaluations like SCROLLS (Shaham et al., 2022b). The results in Table 2 show that recurrent models generally maintain performance beyond their training context (except for Mamba-7b) while transformers (without modification) do not. However, Table 2 also demonstrates that simple linear scaling of the rotary positional embedding (Peng et al., 2023b; emozilla, 2023) can allow for context

scaling beyond the window used for training a given transformer model, effectively nullifying the performance edge of these linear models. Furthermore, transformers generally outperform linear models at their maximum training context length. Further research is needed into extending linear models to long-context inference to take full advantage of the lower inference cost relative to vanilla transformers.

**Limitations.** Since our method relies on initializing with strong pre-trained transformers, our models inherit any of the biases and weaknesses of their base models. Additionally, models that are already instruct-tuned do not linearize as well as base models. Our models suffer from poor performance on MMLU which requires in-context learning (5-shot), a weakness of linear models (Akyürek et al., 2024). We leave the investigation of these weaknesses of linear models to future work and hope that our proposed uptraining approach can help facilitate and accelerate the research in this area.

## 5 Related Work

**Linear Transformers.** The linear transformers introduced in Katharopoulos et al. (2020) lagged behind vanilla transformers in downstream performance, and subsequent architectures such as TransNormer (Qin et al., 2022a) and RetNet (Sun et al., 2023) narrow the gap, but do not demonstrate competitive results with modern transformers at scale. RWKV (Peng et al., 2023a), a linear transformer that takes inspiration from LSTM (Hochreiter & Schmidhuber, 1997), is competitive with compute-matched transformer-based models, but lags behind on a number of NLU benchmarks. Griffin (De et al., 2024) is a concurrent model that takes a hybrid approach, combining a sliding window with linear attention shows impressive performance relative to vanilla transformers, but is trained on a high-quality proprietary dataset.

Another thread in the literature focuses on efficient attention alternatives (Performers (Choromanski et al., 2020), Cosformer (Qin et al., 2022b), LUNA (Ma et al., 2021), RFA (Peng et al., 2021), Attention-free Transformer (Zhai et al., 2021)). All of these approaches sacrifice performances for efficiency. Efficiency improvements for vanilla transformers have narrowed the capabilities gap between vanilla and linear transformers. The KV cache (Pope et al., 2023) greatly narrows the inference efficiency gap between linear and vanilla transformers. RingAttention (Liu et al., 2023) allows for very long context scaling of vanilla attention without approximation. Other methods such as HGRN (Qin et al., 2023), HGRN-2 (Qin et al., 2024b), and FlashLinearAttention (Yang et al., 2024) aim to further improve the linear architecture (throughput, memory, performance, etc.) by using gated attention and other such optimization tricks.

**State Space Models.** State-space models (SSMs) such as H3 (Dao et al., 2022), Hyena (Poli et al., 2023), and Mamba (Gu & Dao, 2023) are recent alternatives to vanilla transformers, combining the strengths of convolutional and recurrent models with efficient hardware implementations. Instead of parallelizing training over the sequence, they produce an efficient way to train the sequential RNN. While these models are competitive with vanilla transformers on some tasks, we show that SSMs share the limitations of linear transformers on several in-context learning and long-context tasks.

**Uptraining Linear Transformers.** Hedgehog (Zhang et al., 2024) builds on the work of Kasai et al. (2021), identifying three different ways of training linear transformers – from scratch, uptraining quadratic transformers for a specific task, and uptraining generally. The authors focus on the first two, and we focus on the third. Moreover, they aim at approximating the softmax attention matrices with linear alternatives. In this work, we do not aim to approximate softmax attention, we replace it with a linear alternative (see ablation above and appendix A). Their method is only tested for smaller scale models and with parameter-efficient fine-tuning for larger models, but presents challenges for scaling for two reasons: (1) their training strategy involves comparing full attention matrices which

is computationally expensive, and not feasible for full fine-tuning of large models with long sequences and (2) their method also inherits the gradient instabilities of linear transformers studied in Sun et al. (2023), while our normalization setup leads to stable uptraining of large models. In addition, Mao (2022) investigates various update rule configurations for uptraining and propose decaying fast weights. Concurrent work such as Chen et al. (2024) and Choi (2024) also try to convert vanilla transformers into linear models, albeit with different uptraining methodologies and varying degrees of success. Notably, DiJiang (Chen et al., 2024) employs a novel Frequency Domain Kernelization technique to conduct the linearization, allowing them to uptrain a 7B model on 40B tokens.

## 6 Conclusion

We introduced SUPRA, a technique for converting large-scale pre-trained softmax transformers into recurrent neural networks, enabling the study of the strengths and limitations of recurrent models at scale with minimal compute cost. Compared to pre-training linear models from scratch, the SUPRA strategy produces competitive models comparable to the best available recurrent LLMs (RWKV and Mamba) at the 7B scale.

We identify the strengths of linear models on standard NLU benchmarks but also the enduring limitations on in-context (i.e. MMLU) and long-context (NarrativeQA, Qasper) tasks, showing that linearized models do not inherit these capabilities from the base softmax transformers.

One possible path to rectifying these limitations is explicitly training for in-context learning (Akyürek et al., 2024). We leave explorations of specialized and instruct data in the context of linear transformers to future work. More sophisticated gating mechanisms as in in Peng et al. (2023a) and De et al. (2024) are promising alternatives to our simple linearization. Using our uptraining method would greatly reduce the necessary time and cost of such experimentation.

# 7 Reproducibility

**Codebase** We train our linear models using our fork of OpenLM (Gururangan et al., 2023) that we modify to include a linear attention function (printed below). We use Lightning Attention 2 (Qin et al., 2024a) that offers a fast Triton (Tillet et al., 2019) kernel for linear attention computation. Evaluations are done with the Eleuther evaluation harness (Gao et al., 2023).

**Data** We train and uptrain models on RefinedWeb (Penedo et al., 2023)(with 2 epochs for our Mamba training), which we tokenize with the pre-trained model's tokenizers. When training from scratch, we used the GPT-NeoX-20B (Black et al., 2022) tokenizer. We tokenize with sequence packing and use a default sequence length of 2048.

**Hyperparameters** We use square matrices with biases for the linear layers in the kernel $\phi$ functions to keep the same feature dimension in the queries and keys. We use the same kernel, with the same weights for both queries and keys and apply a ReLU activation. We use 1000 steps of linear learning rate warmup and a cosine learning rate decay from $3e^{-5}$ to $1e^{-5}$ for our 7B uptrainings and from $3e^{-4}$ to $1e^{-5}$ for our 1B uptrainings and for trainings from scratch. We use the Adam optimizer with $\beta_1 = 0.9$ and $\beta_2 = 0.95$. We trained our models with mini-batches totaling 2M tokens each. Our default RoPE frequency uses the Llama value of $10^4$. For longer sequence lengths, we use a RoPE frequency of $10^6$.

**Training** Depending on the model size and the availability, we use from 4 to 32 nodes of 8 GPUs Nvidia H100 with Pytorch FSDP. We use a mixed precision strategy from OpenLM that automatically selects between bfloat 16 and float 32 for different operations. A linear 7B parameter model uptraining throughput is around 4300 tokens per second per GPU.

**Models** Our results can be reproduced by following the same training recipe or using the model weights that we release: Mistral-SUPRA and Mamba-7b.

```python
def linear_attn_func(q, k, v, qk_scale: float, use_decay: bool = True,
    normalize: bool = False) -> torch.Tensor:
    """
    Args:
        q: queries, shape (batch_size, num_heads, seq_len, dim_qk)
        k: keys, shape (batch_size, num_heads, seq_len, dim_qk)
        v: values, shape (batch_size, num_heads, seq_len, dim_v)
        qk_scale: scale factor for queries and keys
    """
    h = q.shape[1]
    if use_decay:
        s = slope_tensor(h, q.device, q.dtype)
    else:
        s = no_slope_tensor(h, q.device, q.dtype)

    output = lightning_attn_ops(q, k * qk_scale, v, s)
    if normalize:
        norm = torch.clamp_min(
                    torch.einsum(
                        "nhld, nhld->nhl", q, k.cumsum(2) * qk_scale
                    ), 1e-6
                )
        return output / norm.unsqueeze(-1)
    else:
        return output
```

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

# A Attention Approximation

In this section we investigate whether our up-training procedure leads to linear attention that approximates the softmax attention from the base model, as might be expected.

There are many possible ways to compare attention matrices. Moreover, some architecture changes such as attention decay and lack of normalization in the linear attention make a meaningful comparison difficult. We represent non-normalized comparisons in Figure 4. It represents the cosine similarities and singular value distances between the attention matrices at every layer and for every head of the Mistral model compared with our Mistral-SUPRA. Each pixel of these images is a scalar similarity measure between two matrices represented by a color scale. In Figure 4, we see large differences between the matrices.

Since we removed the attention matrix normalization and replaced it with a LayerNorm Ba et al. (2016), we want to compare normalized attention matrices instead. We divide each line of the matrix by the absolute value of the sum of its elements such that the softmax attention matrix is unaffected and the linear attention matrix is normalized. In Figure 5, we see significantly higher between most matrices with some exceptions. These observations indicate that the linear attention matrices derived from SUPRA are *not an approximation* of the softmax matrices.

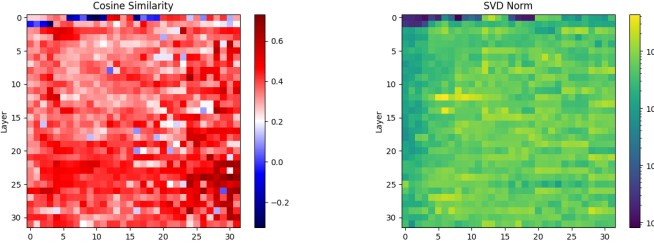

Figure 4: Representation of the cosine similarity and the distance between the singular values of the softmax attention matrices compared to the SUPRA attention matrices.

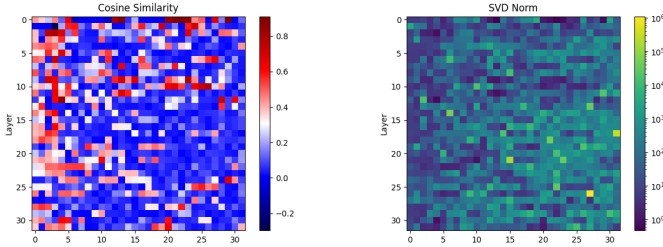

Figure 5: Representation of the cosine similarity and the distance between the singular values of the *normalized* softmax attention matrices compared to the *normalized* SUPRA attention matrices.

