# OpenReview forum: "Linearizing Large Language Models"
_colmweb.org/COLM/2024/Conference — COLM_

### Official Review · Reviewer_XQ7B · 2024-04-18

**Rating:** 6
**Confidence:** 4
**Ethics Flag:** 1

**Summary:**

This paper follows the previous line of works in finetuning softmax attention LMs to linear attention ones. The benefit is (1) to reduce the cost of training from scratch: only need 10% training tokens to achieve similar performance, as shown in this paper; (2) to reduce the inference cost by transforming to the recurrent mode during inference.

This paper adopts many advanced linear attention techniques to enhance the performance, for example using decays and output normalization. The authors give an important observation that output normalization is crucial for stable finetuning, while directly adopting the methods from T2R will lead to inferior results.

More interestingly, this work also demonstrates some drawbacks of linear models, for example the incapability in long-text modeling and incontext learning.

**Questions To Authors:**

Questions and Suggestions:

1. In table1, it seems that continual training for more tokens could bridge the gap between linear and softmax attention. It would be interesting to continually train on a small model (e.g., 1B TinyLLaMa) for more tokens to see, to what extent, linear attention could bridge the performance gap to softmax attention.

2. In table3, why do you only report the results of HellaSwag and ARC? Do you have a specific reason? What about other metrics?

3. Could you release an an anonymous huggingface repo for the 7B mamba model?

4. It would be interesting to use the attention distillation loss proposed in Hedgehog for learning the feature map?

5. It would be interesting to try data-dependent decays proposed in Gated Linear Attention. https://github.com/sustcsonglin/flash-linear-attention



Missing reference: (although some are released after the submission deadline, it is beneficial to discuss in the next iteration):

- linear attention with data-dependent decays [1, 2, 5].
- Fine-tuning attention-based LMs into linear attention [1, 3, 4].

[1] Fine-Tuning Pre-trained Transformers into Decaying Fast Weights. https://aclanthology.org/2022.emnlp-main.697.pdf

[2] Gated Linear Attention Transformers with Hardware-Efficient Training. https://arxiv.org/abs/2312.06635

[3] Cross-Architecture Transfer Learning for Linear-Cost Inference Transformers. https://arxiv.org/abs/2404.02684

[4] DiJiang: Efficient Large Language Models through Compact Kernelization. https://arxiv.org/abs/2403.19928

[5] HGRN2: Gated Linear RNNs with State Expansion https://arxiv.org/abs/2404.07904

**Reasons To Accept:**

- Although the techniques are not new (reason to reject 1),  the large-scale empirical verification of these techniques is important to the community. This paper also presents some interesting empirical results, revealing the incapability of linear attention/RNNs in long text modeling and in-context learning.

**Reasons To Reject:**

1. No new techniques are proposed. All methods are proposed before, including incorporating decaying factors, applying output normalization and trainable MLP feature map, etc. The main contribution to me is re-emphasizing the importance of output normalization in the large-scale funetuning setting.

2. The perplexity values should be reported for all experiments.  Perplexity is still one of the most important metrics.

3. I found the paper lacking an important baseline: continually train the original model on the same data to see if MMLU declines a lot. If so, this is not the issue of linear Transformers, but the issue of the training data itself. This is a very important baseline. I will increase the score if the author can answer this question in depth.

---

> ### Author Rebuttal · Authors · 2024-05-31
>
> Thanks for taking the time to give detailed feedback and suggestions! We agree with them and feel that incorporating these comments will make our paper stronger. Due to length limitations, we kept our responses short and had to omit some questions. We are happy to provide more details during the discussion period.
>
> ---
>
> > I found the paper lacking an important baseline: continually train the original model [...]
>
> Good point! We include these results below. Here, we uptrain Llama2-7B on 20B tokens of RefinedWeb, which is the same setting used to uptrain Llama2-SUPRA.
> |Model|New tokens|HS|PIQA|WG|ARC-E|ARC-C|MMLU|Average|
> |--|--|--|--|--|--|--|--|--|
> |Llama2 (base)|-|76.0|79.1|69.1|76.3|46.3|45.9|65.4|
> |Llama2|20B|76.3|78.8|70.1|76.8|46.2|43.8|65.3|
> |Llama2-SUPRA|20B|71.8|78.6|65.8|71.1|39.5|24.9|58.6|
>
> All the evaluations are preserved with continual training except for MMLU which shows a slight drop when uptraining the softmax Llama2 (45.8->43.8). The MMLU performance drop is much wider for Llama2-SUPRA (45.8->24.9).
>
> This tells us that the MMLU performance drop can be attributed to both the architecture (linearization procedure) and the uptraining process (data, hyperparameters), with the linearization procedure being the bigger contributor.
>
> > The perplexity values should be reported for all experiments.
>
> Previous works have relied heavily on perplexity to report good performance of linear models at context extension. We have grown skeptical about the pertinence of these results when evaluating long-context tasks and noticing discrepancies. Nonetheless, we agree that this is one of the most important metrics and that including these results would strengthen our paper.
>
> Below we compare the average cross entropy as a function of the token position in the sequence for Llama2-Uptrain-10B vs Llama2-SUPRA on a val set (C4 subset).
>
> |Model|8|32|128|256|512|1024|2048|
> |-|-|-|-|-|-|-|-|
> |Llama2 (base)|2.76|2.70|2.04|1.94|1.85|1.82|1.78|
> |Llama2-Continued-20b|2.76|2.69|2.05|1.94|1.84|1.82|1.79|
> |Llama2-SUPRA|2.86|2.79|2.18|2.07|2.02|2.01|2.00|
>
> This adds further evidence to our claim that linear uptrains perform poorly at higher context lengths – in early tokens, the gap between Llama2-uptrain and Llama2-SUPRA is small, but as we go to later tokens, the loss for Llama2-uptrain continues to decrease while the loss for Llama2-SUPRA stays high.
>
> In the final version, we will include perplexity values for all our other experiments and models.

---

> ### Author Response · Authors · 2024-06-03
> **additional answers**
>
> We would like to thank you for increasing the score after our first answer. For completeness, we wanted to add answers to the other questions that you raised. We are happy to discuss more if you have any additional questions.
>
> > 5. It would be interesting to try data-dependent decays proposed in Gated Linear Attention.
>
> We actually tried this previously and noticed only a slight difference.
> - 1B SUPRA; 10B RW tokens; HellaSwag = 57.0
> - 1B GLA; 10B RW tokens; HellaSwag = 57.6
>
> We believe the data dependent decay suffers from the same problem as fixed decay, which we mention in the paper (decays too fast and does not allow long context attention). We are continuing to look into this for future work, but this would require a thorough study of gating mechanisms and is out of the scope of this current work.
>
> > 1. In table1, it seems that continual training for more tokens could bridge the gap between linear and softmax attention.
>
> As training is pushed from 20B tokens to 70B tokens, the results do improve (Table 1). They would probably improve further with even longer training. However, training for more tokens would bring the cost closer to pretraining. In our experiments, our approach seems most relevant in the context of uptraining with a modest compute budget. With more available compute, it is likely that other approaches such as Mamba would yield better results at lower cost and would therefore be preferable.
>
> > 2. In table3, why do you only report the results of HellaSwag and ARC? Do you have a specific reason? What about other metrics?
>
> For Table 3, we only considered 1B models for ablations. Since 1B models are weaker than the 7B models in Table 1, we opted to evaluate on easier test sets such as HellaSwag and ARC. Evaluating on a difficult set like MMLU would not be very meaningful because the results are just close to random guessing.
>
> > 4. It would be interesting to use the attention distillation loss proposed in Hedgehog for learning the feature map?
>
> Thank you for bringing this up! We were aware of the Hedgehog paper and in fact cited it in our related work. However, **there is no released code** beyond the pseudocode attached to the paper. This made it time-consuming to implement and test, but we agree that it is an interesting baseline and will try to include it in our camera ready version.
>
> More specifically, a naive implementation of their approach would require two full computations of the attention matrices to compute the loss that the authors introduce. This is known to be memory intensive and we would not be able to train at scale with this approach without an involved implementation of memory efficient attention with an added attention loss.
>
> Furthermore, in our Appendix, we show that our trained SUPRA matrices do not actually approximate the softmax matrices. The RetNet paper shows how the normalization of the attention factor that allows to approximate the softmax can lead to instabilities. When uptraining 1B models, we did experience these instabilities in practice. Thus, while we lack the experiments that would give a definitive answer to your question, we believe that the Hedgehog approach of trying to very closely approximate softmax may not be the correct approach for uptraining such linear models at scale.
>
> Unfortunately, the Hedgehog paper does not publish evaluations on the common benchmarks like HellaSwag, ARC, and PIQA which does not allow a direct comparison of our respective results.
>
>
> > 3. Could you release an anonymous huggingface repo for the 7B mamba model?
>
> Yes, we plan to release our models and code together with this submission. For the 7B mamba, we have created an anonymous model on Hugging Face https://huggingface.co/mambacolm/mambacolm
>
>
> Thanks as well for including the missing references! We will include these in the final version of the paper.

---

> > ### Comment · Reviewer_XQ7B · 2024-06-06
> >
> > Thank you for these results! I have two comments:
> >
> > - 1B SUPRA vs. 1B GLA: Do you have any results beyond the HellaSwag dataset? I don’t consider a 0.6 difference to be slight.  It might be worth investigating whether training on more tokens offers an improvement over SUPRA's method with data-independent decays.
> >
> > - Regarding Computational Resources: I understand the limitation regarding computational resources. However, since you were able to train the 7B Mamba model on 1T tokens as a baseline, it might be more fruitful to explore pushing this approach to its limits

---

> > ### Author Response · Authors · 2024-06-07
> >
> > Thank you for your interest and for your answer. We agree that a 0.6 difference is not insignificant and our further results below do suggest that GLA is indeed a stronger approach than the fixed decay used in SUPRA. It is worth studying further for long context and at larger scale but we believe this to be a broader study.
> >
> > |Model|Size|Tokens|HellaSwag|ARC-E|ARC-C|
> > |-|-|-|-|-|-|
> > |SUPRA|1B|(1.6T)+10B|57.0|62.4|31.6|
> > |SUPRA GLA|1B|(1.6T)+10B|57.6|64.3|33.1|
> >
> > Additionally, we would like to share our perplexity evaluation as a function of context length to show why we believe that GLA improves the model overall but does not solve the long-context modeling capability issue of linear models. While showing a lower loss with GLA, linear model performance (including GLA) tend to plateau as the context is extended.
> >
> > |Model|8|32|128|256|512|1024|2048|
> > |-|-|-|-|-|-|-|-|
> > |SUPRA GLA 1b c4|3.75|3.15|2.91|2.68|2.55|2.71|2.58|
> > |SUPRA 1b c4|3.75|3.18|2.94|2.71|2.59|2.76|2.68|
> > |Softmax 1b c4|3.69|3.06|2.8|2.48|2.36|2.56|2.38|

---

### Official Review · Reviewer_JrUb · 2024-04-30

**Rating:** 7
**Confidence:** 4
**Ethics Flag:** 1

**Summary:**

This paper proposes Scalable UPtraining for Recurrent Attention (SUPRA), a method to convert existing pretrained transformers into a linear transformer that is much more efficient in terms of data and inference. The authors conducted extensive analyses to validate their proposed methods. The results show that SUPRA can maintain much of the performance of the original transformers, though the performance is only competitive with existing RNNs and below original transformers.

**Questions To Authors:**

- In table 2, it is mentioned that *"†These results were evaluated on subset of the first samples and will be updated"*. I don't think this is a good practice that should be encouraged. It is the authors' responsibility to make sure that the results are complete for review. Otherwise, incomplete evaluations might cause misleading conclusions.

**Reasons To Accept:**

- This is a very interesting paper exploring the intersection between RNNs and transformers. Linearizing transformers seems to be a pretty interesting and simple approach to achieve a good compromise between efficiency and performance. While the linearized model is not able to outperform the orignal transformers and existing RNNs for now, it does offer an alternative to existing approaches. Personally, I think the community should encourage such exlorations in addition to SOTA chasing.
- This paper raises some new interesting questions, especially about how linear transformers are competitive with transformers in some tasks but lag behind in others (in-context learning, instruction-tuning).
- The authors have made good efforts to ensure the reproducibility of this paper.

**Reasons To Reject:**

- Kernel choice. The paper only looks at a very specific linear kernel but I would appreciate more justifications. The chosen kernel has a non-negligible impact on the final performance (Tsai et al., 2019; Arora et al., 2024 - the latter is too new so it's not a problem for not discussing it). The proposed method has made some incremental improvement to T2R but more in-depth analysis of the impact of kernel can greatly improve this paper.

**References**
- Tsai, Y. H. H., Bai, S., Yamada, M., Morency, L. P., & Salakhutdinov, R. (2019). Transformer dissection: a unified understanding of transformer's attention via the lens of kernel. arXiv preprint arXiv:1908.11775.
- Arora, S., Eyuboglu, S., Zhang, M., Timalsina, A., Alberti, S., Zinsley, D., ... & Ré, C. (2024). Simple linear attention language models balance the recall-throughput tradeoff. arXiv preprint arXiv:2402.18668.**

---

> ### Author Rebuttal · Authors · 2024-05-31
>
> Thank you for your positive review! We aimed to provide a comprehensive view of the linearization process and examine its viability as an alternative to pretraining linear models. We are glad you appreciated our work and specifically the comparison of the strengths and limitations compared to softmax transformers.
>
> ---
>
> **Kernel Choice**:
> The kernel choice was ablated in Table 3. We tried a few different things but the highest performance was the RetNet-like kernel. We will also clarify the choices of the kernel and add further ablations in our final version. Thank you for the suggestion!
>
> **Regarding Table 2**:
> We agree with the reviewer that evaluating on a subset is not ideal. We apologize for underestimating the time it would take to compute the 16k results and not filling the full test at submission time. For this specific case, evaluating on the full evaluation set gave us very similar results and insights as our reported results. We will include complete results in the final version.

---

> > ### Author Response · Authors · 2024-06-04
> > **Additional kernel experiments**
> >
> > We would like to complete our answer with the additional kernel experiments that we have done. Please refer to [our answer to reviewer JkbR](https://openreview.net/forum?id=soGxskHGox&noteId=g6AZlJ3S2v) that has expressed a similar interest in a more thorough ablation of the kernel choice.

---

### Official Review · Reviewer_TZps · 2024-05-11

**Rating:** 6
**Confidence:** 4
**Ethics Flag:** 1

**Summary:**

This paper explores a way to fine-tune pre-trained Transformers into RNNs, which benefit from efficient inference. It is based on the work of Kasai et al. (EMNLP 2021), with improved model components and new experiments applied to a larger scale, and to compare with some recent pre-trained RNN models.

**Reasons To Accept:**

Linear Transformers is an intriguing topic. This paper is timely with a solid contribution of some new experiment results.

**Reasons To Reject:**

There is not much highlight in the proposed method. It tweaks the model of Kasai et al. (EMNLP 2021) by switching to components that empirically turn out to be better in recent years, which I believe would improve performance but there is no conceptually new idea. In the Introduction, the authors compare this work with Kasai et al. and claim that "We take a different approach: rather than approximate softmax attention, we instead replace it with a linear kernel and a normalization strategy to fine-tune the most performant LLMs into RNNs. We take advantage of models trained on high-quality, proprietary datasets..." But Kasai et al. is actually doing the same thing -- it didn't use the GroupNorm component as in this paper and didn't apply to pre-trained models of 7B scale, but that doesn't mean it cannot -- the experiments conducted in this paper are not newly enabled because of the method proposed in this paper.

Moreover, the ablation test shown in Table 2 does not seem to support that the proposed method has any specific strength.

Overall, the narration of this paper is messy. It indicates that the authors lack a well-polished story.

One interesting result from the experiments, is that even at the very large 7B scale, fine-tuning a pre-trained Transformer into an RNN can immediately decrease the performance of some tasks -- more detailed investigation into this issue might be a good direction to explore in the future.

---

> ### Author Rebuttal · Authors · 2024-05-31
>
> Thanks for your review and for appreciating our experimental results. Our goal was to find a better way to linearize LLMs while still preserving strong performance. We focused on this last aspect as we found that the full picture of evaluations of linear models was missing from the literature.
>
> ---
>
> > Kasai et al. is actually doing the same thing [...] The experiments conducted in this paper are not newly enabled because of the method proposed.
>
> While our method for linearization is based on T2R (Kasai et al), both our experiments and our discussion significantly differ from that work. We argue that our paper, while working in the same framework as T2R, **enables scaling**. Some design choices may seem obvious in hindsight, but scaling is a nontrivial task. In Table 3, we show that naively scaling up T2R is unstable and fails. We did the work of testing different design choices and finding ways to outperform most concurrent linear models with a lower training budget (Table 1).
>
> In addition, we argue that the value of our work lies not only in architectural elements but also in our thorough experimental evaluation. For instance, we explore performance of transformers and linear models (not just our models but also others) on long-context tasks and tasks like MMLU. This is something that previous papers like RetNet or T2R did not address.
>
> > The ablation test shown in Table 2 does not seem to support that the proposed method has any specific strength.
>
> Our goal is not to produce linear models that match softmax transformer performance. Rather, we aim to provide a holistic view of the strengths/weaknesses of our linearization process. As you mentioned, Table 2 shows that our models do not perform well on long-context. At the same time, Table 2 also shows that linear models in general (Mamba, RWKV) do not perform well on long-context, and that is something that we wanted to highlight in our paper. Many previous works imply that linear models perform well on long-context tasks. Thus, as noted by other reviewers, we believe Table 2 to be a strong experimental result: not in the sense that it shows any advantage of our method, but because it is a new result clearly identifying the limitations of linear transformers.
>
> Having said that, we would like to point out that our model outperforms other linear models like RWKV on multiple benchmarks (HellaSwag, PIQA) with much fewer training tokens, which we highlight as one of the strengths of our proposed method.

---

### Official Review · Reviewer_JkbR · 2024-05-13

**Rating:** 6
**Confidence:** 5
**Ethics Flag:** 1

**Summary:**

This paper studies an interesting question of finetuning a standard attention-based LLM into a sub-quadratic model (i.e., linear attention models in this work). This paper tried to address several technical problems: 1) which sub-quadratic architecture is more compatible with finetuning, 2) choice of decay, kernel functions and normalization, 3) whether to fine-tune in a two-stage way.  Overall, this paper reveals the potential of finetuing open-source LMs into their sub-quadratic versions as an alterative to training one from scratch.

**Questions To Authors:**

* I wonder whether the kernel function of relu (following T2R) is really necessary, compared with using an identify function as in RetNet.
* it would be better to compare with the following baseline [1]

[1] The Hedgehog & the Porcupine: Expressive Linear Attentions with Softmax Mimicry

**Reasons To Accept:**

An interesting work showing the potential of uptraining current open-source LMs (e.g., llama and mistral) into linear models. It also reveals several limitations of uptrained models, which are useful for future research: they are harder to benefit from longer-context training, and still suffer from poor few-shot learning performance.

**Reasons To Reject:**

In terms of technical novelty, the contribution of this paper is rather limited: the architecture for uptraining basically follows RetNet (including incorporation of RoPE and decaying factor).

---

> ### Author Rebuttal · Authors · 2024-05-31
>
> Thank you for your detailed review of our paper. We wanted to provide a holistic view of the linear uptraining process and highlight both its strengths and limitations. We appreciate your acknowledgment of the interesting aspects of our work and the potential it shows for uptraining current open-source LMs into linear models.
>
> ---
>
> > In terms of technical novelty, the contribution of this paper is rather limited: the architecture for uptraining basically follows RetNet (including incorporation of RoPE and decaying factor).
>
> Although our kernel closely follows the design in RetNet, **the way we train the model is very different**. We demonstrate that by uptraining pretrained models, we can achieve RetNet-like performance much more quickly. This is reflected in our results: with the same number of tokens (100B), our model significantly outperforms RetNet (e.g. HellaSwag 60.7 vs **77.1**). Furthermore, RetNet experiments were limited to 2048 context training; we show the challenges of scaling the approach further.
>
> In addition, we argue that the value of our paper lies not only in the introduction of new architectural elements but also in the thorough experimental evaluation we conducted. For instance, we explore the performance of transformers and linear models (not just our SUPRA models but also others) on long-context benchmarks and on benchmarks like MMLU. This is something that previous papers such as RetNet or T2R did not address.
>
>
> > I wonder whether the kernel function of relu (following T2R) is really necessary, compared with using an identify function as in RetNet.
>
> We chose the ReLU kernel function because it allows non-negative attention factors which is desirable given that softmax attention is also non-negative.
>
> > It would be better to compare with the following baseline
>
> Thank you for bringing this up! We were aware of the Hedgehog paper and in fact cited it in our related work. However, **there is no released code** beyond the pseudocode attached to the paper. This made it time-consuming to implement and test, but we agree that it is an interesting baseline and will try to include it in our camera ready version.
>
> Unfortunately, the Hedgehog paper also does not publish evaluations on the common benchmarks like HellaSwag, ARC, and PIQA which does not allow a direct comparison of our respective results.

---

> > ### Comment · Reviewer_JkbR · 2024-05-31
> >
> > Thanks for the reply.
> >
> > I understand that the main contributions lie in the new training procedure, and agree that the results on MMLU and long-context benchmarks are valuable.
> >
> > Re relu vs identity, I'm wondering whether having non-negativity is really necessary from the empirical perspective (because linear attentions are not strictly mimicking softmax attention anyway.

---

> > > ### Author Response · Authors · 2024-06-04
> > > **Kernel experiments**
> > >
> > > Thank you for your answer. We agree that adding more experiments with different kernel activation functions completes Table 3 nicely, strengthens the paper, and should answer your question. We uptrained combinations of kernel variants for 10B tokens from a pre-trained 1B parameters model:
> > >
> > > The "no embed" models remove the additional fully connected layer that we added. Removing this layer significantly degrades the results. The "1+elu" models replace the ReLU activation with
> > > $f(x)=1+elu(x)$, as used in  [Transformers are RNNs](https://arxiv.org/abs/2006.16236), which is simply a smooth version of the ReLU function. The "no activation" models do not use any activation function. The bias added by the fully connected layer of the kernel is important for producing good results. However, the activation itself has little impact on performance. That being said, the results are slightly better with a non-linear activation than without.
> > >
> > > |Model|Size|Tokens|HellaSwag|ARC-E|ARC-C|
> > > |-|-|-|-|-|-|
> > > |SUPRA|1B|(1.6T)+10B|57.0|62.4|31.6|
> > > |SUPRA-no activation|1B|(1.6T)+10B|55.9|61.6|29.8|
> > > |SUPRA-no embed-no activation|1B|(1.6T)+10B|36.5|48.5|23.7|
> > > |SUPRA-1+elu|1B|(1.6T)+10B|56.5|64.0|31.1|
> > > |SUPRA-no embed-1+elu|1B|(1.6T)+10B|49.9|57.4|30.6|

---

### Decision · Program_Chairs · 2024-07-10

**Decision:**

Accept

**Comment:**

This paper proposes Scalable UPtraining for Recurrent Attention (SUPRA), a method to uptrain existing transformer-based LLMs into RNNs (linear attention) with limited compute. The novelty (3/4 reviewers critique is too be *too limited*) isn't in the architecture (that's mostly taken from previous work), but rather in the empirical work on how to *train* such an architecture (with some updated components) and identifying the limitations of the resulting model. Perhaps it is the observations of limitations of the new model in terms of in-context learning that is the most interesting/original of the work, but it might not be enough to justify acceptance.

Pros:
- Practically useful observations on how to uptrain an attention-based LLM into an RNN.
- Interesting discussion on the limitations of such a model in terms of in-context learning.

Cons:
- Lack of novelty (3/4 reviewers) since mostly reuses an architecture proposed in previous work.
- Messy narrative (TZps) and some doubts about the reported results and baselines (TZps, XQ7B).